# Sparse Polynomial Chaos Expansion for Uncertainty Quantification of Composite Cylindrical Shell with Geometrical and Material Uncertainty

**Ming Chen [1,2,*], Xinhu Zhang [1,2,*], Kechun Shen [1,2,3] and Guang Pan [1,2]**

1   School of Marine Science and Technology, Northwestern Polytechnical University, Xi'an 710072, China; shenkechun@126.com (K.S.); panguang@nwpu.edu.cn (G.P.)
2   Key Laboratory for Unmanned Underwater Vehicle, Northwestern Polytechnical University, Xi'an 710072, China
3   Structural Ceramics and Composites Engineering Research Center, Shanghai Institute of Ceramics, Chinese Academy of Sciences, Shanghai 200050, China
*   Correspondence: xiuzhandao@126.com (M.C.); xinhu_zhang@126.com (X.Z.)

**Abstract:** The geometrical dimensions and mechanical properties of composite materials exhibit inherent variation and uncertainty in practical engineering. Uncertainties in geometrical dimensions and mechanical properties propagate to the structural performance of composite cylindrical shells under hydrostatic pressure. However, traditional methods for quantification of uncertainty, such as Monte Carlo simulation and the response surface method, are either time consuming with low convergence rates or unable to deal with high-dimensional problems. In this study, the quantification of the high-dimensional uncertainty of the critical buckling pressure of a composite cylindrical shell with geometrical and material uncertainties was investigated by means of sparse polynomial chaos expansion (PCE). With limited design samples, sparse PCE was built and validated for predictive accuracy. Statistical moments (mean and standard deviation) and global sensitivity analysis results were obtained based on the sparse PCE. The mean and standard deviation of critical buckling pressure were 3.5777 MPa and 0.3149 MPa, with a coefficient of variation of 8.801%. Global sensitivity analysis results from Sobol' indices and the Morris method showed that the uncertainty of longitudinal modulus has a massive influence on the critical buckling pressure of composite cylindrical shell, whereas the uncertainties of transverse modulus, shear modulus, and Poisson's ratio have a weak influence. When the coefficient of variation of ply thickness and orientation angle does not surpass 2%, the uncertainties of ply thickness and orientation angle have a weak influence on the critical buckling pressure. The study shows that the sparse PCE is effective at resolving the problem of high-dimensional uncertainty quantification of composite cylindrical shell with geometrical and material uncertainty.

**Keywords:** sparse polynomial chaos expansion; high dimensionality; composite material; geometrical uncertainty; material uncertainty

## 1. Introduction

The past several decades have witnessed a great increase in the number of underwater vehicles manufactured using composite materials for both commercial and military purposes [1]. Light weight is of great significance for underwater vehicle applications [2,3]. Therefore, high-performance composite materials have been increasingly applied due to their high specific strength–weight ratio and stiffness–weight ratio, excellent design flexibility, and corrosion resistance [4–6].

Cylindrical shells are a crucial component of underwater vehicles due to their resistance to external hydrostatic pressure. For cylindrical shells with large radius–thickness

ratios, buckling induced by external hydrostatic pressure dominates the structural failure form when external hydrostatic pressure surpasses the critical load [7–10].

Composite cylindrical shells consist of layers with various orientation angles and thicknesses, and even with different fibers, and can fulfill the requirement of desired strength and stiffness. As an effective tool for predicting the critical buckling load of composite cylindrical shells, numerical solutions have been adopted by researchers to investigate the effect of length [11], shell thickness [12], orientation angle, and stacking sequence [13–15]. Shen [16] studied the buckling and strain response of filament winding composite cylindrical shells under hydrostatic pressure by numerical analysis and experimental testing. Rezaiee-Pajand and Masoodi [17–19] reviewed the progress of buckling and post-buckling behavior of plates and shells, and performed nonlinear analysis of thin and moderately thick panels of functionally graded material.

Several researchers [20–22] have performed numerical investigations on the influence of manufacturing defects on the buckling load of composite cylindrical shells and conducted experiments. Zhang [23] studied the effects of the ovality and the thickness eccentricity on the collapse of subsea cylindrical pipelines with imperfections. Teixeira [24] conducted a reliability analysis of cylindrical pipelines with local defects. Geometrical variation and uncertainty induced by manufacturing defects have a great influence on structural performance.

Composite materials exhibit variations and uncertainties with respect to their geometrical dimensions and material properties. Geometrical and material uncertainties propagate to the structural performance of composite cylindrical shells under hydrostatic pressure. Cai [25,26] performed reliability-based load and resistance factor design for composite cylindrical shells under hydrostatic pressure, and later conducted a probabilistic analysis by means of Monte Carlo simulation and the response surface method. Ly [27] performed uncertainty quantification on the critical buckling load of columns under axial compression with uncertain random materials. Gerhardt et al. [28] performed numerical and experimental structural instability analysis of composite tubes considering manufacturing parameters and imperfections. Zhou [29] reviewed the stochastic multiscale analysis of FRP composite structures and found that uncertainties at different scales should be considered simultaneously. Considering the uncertainties of winding angle, fiber volume fraction, mechanical and strength properties, Solazzi and Vaccari [30] performed a reliability analysis of a pressure vessel made with steel and carbon and glass fiber composite materials, it was found that weight reduction was reduced greatly for the composite pressure vessel, with the reliability being comparable to that of the vessel made of aluminum and steel. Rafiee [31] implemented Monte Carlo simulation to predict the burst pressure of composite pressure vessels. The results showed that the composite pressure vessel was highly likely to experience burst below the average burst pressure. Hocine et al. [32] employed the maximum stress and Tsai-Wu failure criteria for the failure analysis of composite tubular structures and performed a sensitivity analysis. It was found that winding angle and ply thickness had a significant influence on pressure load. Balokas [33] conducted inverse uncertainty quantification of the transverse tensile response of carbon fiber-reinforced composites, and the results showed that the uncertainty was greatly reduced. Kalfountzos [34] used the probabilistic finite element method to determine the probabilistic buckling response of fiber–metal laminate panels under uniaxial compression and found that realistic structural uncertainties could substantially affect the buckling strength. Yetgin [35] utilized Monte Carlo simulation to investigate the first ply failure and burst pressure of a filament wound pressure vessel taking into account the uncertainty of the material properties and winding angles. The results showed the importance of considering the uncertainties of the material properties and the winding angle when predicting the mechanical performance of composite pressure vessels. On the basis of the literature review, uncertainty analysis concerning long composite tubes and composite pressure vessels subjected to internal pressure has been investigated, while uncertainty analysis for composite cylindrical shells under external hydrostatic pressure has rarely been thoroughly studied. It was also found

that geometrical and material uncertainties are important factors affecting structural performance. Composite cylindrical shells consist of layers with various orientation angles and thicknesses, and the geometrical uncertainties of orientation angle and thickness for each ply are not simultaneously considered due to the increasing dimensionality and complexity.

Composite cylindrical shells are a vital component of underwater vehicles due to their ability to withstand external hydrostatic pressure. However, existing studies on composite cylindrical shells in underwater vehicles have mainly focused on deterministic analysis, without taking into account the inherent uncertainties of geometrical dimensions and material properties associated with composite materials. In fact, geometrical uncertainties (ply thickness and orientation angle) and material uncertainties greatly affect the structural performance of the composite cylindrical shell. Composite cylindrical shells are stacked by ply during the manufacturing process, and considering the geometrical uncertainties of each ply results in increasing dimensionality and complexity. Confronted with a high-dimensional problem, for simplicity, a certain orientation angle (positive or negative values) and total shell thickness are considered, rather than ply thickness, for each ply [25,26,30,34]. Thus, sparse PCE is adopted to resolve the high-dimensional problem while taking into account the uncertainties of ply thickness and orientation angle for each ply.

This paper investigates the effects of uncertainties of material properties and geometrical dimensions on the critical buckling pressure of composite cylindrical shells in underwater vehicles under hydrostatic pressure. The longitudinal elastic modulus, transverse modulus, shear modulus, Poisson's ratio, orientation angle and ply thickness of the composite material are treated as random input variables, and the critical buckling pressure is treated as a random output variable. Random input variables are sampled by the Latin hypercube method for the purpose of design of experiment (DOE), and the critical buckling pressure is evaluated by finite element (FE) analysis. A data-driven sparse polynomial chaos expansion (PCE) for high-dimensional uncertainty quantification of composite cylindrical shell is constructed and validated. Global sensitivity analysis based on the constructed sparse PCE is performed to identify the most influential random variables.

## 2. Finite Element Method

### 2.1. Finite Element Modeling

A composite cylindrical shell for an underwater vehicle subjected to external hydrostatic pressure is investigated. The composite cylindrical shell has a length of 800 mm, a diameter of 324 mm, and a thickness of 6 mm. Abaqus, a commercial program, is used for finite element (FE) analysis. The S4R element is used to model the composite cylindrical shell, as shown in Figure 1. As sufficient and good-quality data are lacking, the values of the variables are determined on the basis of a review of the references, testing, and engineering judgment. The material properties of the carbon fiber–epoxy composite and its statistical information are listed in Table 1. The stacking sequence of the composite cylindrical shell is [90/60/−45/−60/90/0/0/30/60/90] s, with a ply thickness of 0.3 mm; both the orientation angle and the ply thickness are subject to normal distribution with a coefficient of variation of 2%, as presented in Table 2. The right end cover is made of ASTM A36 steel (E = 200 GPa, Nu = 0.26), with a thickness of 20 mm, and is modeled using the C3D8R element. As for constraints, the composite cylindrical shell and the right end cover are tied. The right end cover has greater stiffness and is chosen as the master surface, and the composite cylindrical shell is chosen as the slave surface. As for the boundary conditions for the finite element analysis, the left end of the composite cylindrical shell is clamped, allowing axial displacement at the right end cover. As for the load for the finite element analysis, the composite cylindrical shell is subjected to external uniform pressure.

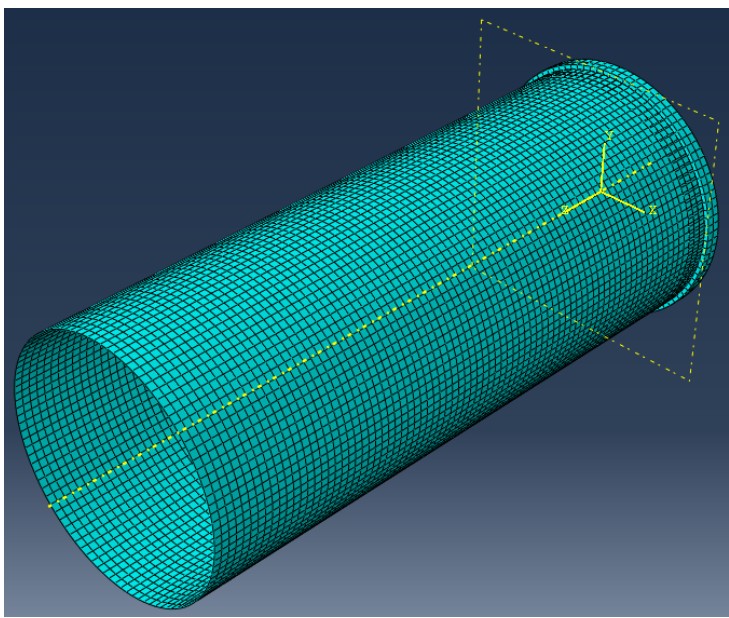

**Figure 1.** Finite element mesh of the composite cylindrical shell.

**Table 1.** Statistical information of the material properties of the carbon fiber–epoxy composite material.

| Symbol | Unit | Mean | Coefficient of Variation | Distribution |
|--------|------|------|--------------------------|--------------|
| E1 | GPa | 121 | 10% | Normal |
| E2 | GPa | 8.6 | 6% | Normal |
| E3 | GPa | 8.6 | 6% | Normal |
| Nu12 | - | 0.253 | 8% | Normal |
| Nu13 | - | 0.253 | 8% | Normal |
| Nu23 | - | 0.421 | 8% | Normal |
| G12 | GPa | 3.35 | 8% | Normal |
| G13 | GPa | 3.35 | 8% | Normal |
| G23 | GPa | 2.68 | 8% | Normal |

**Table 2.** Statistical information of the geometrical dimensions of the composite cylindrical shell.

| Property | Symbol | Unit | Mean | Coefficient of Variation | Distribution |
|----------|--------|------|------|--------------------------|--------------|
| Ply thickness | Ti (i = 1, 2, . . . , 10) | mm | 0.3 | 2% | Normal |
| Orientation angle | Ai (i = 1, 2, . . . , 10) | degree | θ | 2% | Normal |

The modeling strategy described above was adopted for the finite element analysis of 12 models in Ref. [36]. The stacking sequences of the 12 models were [±30/90] FW, [±45/90] FW, [±60/90] FW, with four models for each stacking sequence. The average critical buckling pressures for the stacking sequences [±30/90] FW, [±45/90] FW, [±60/90] FW in Ref. [36] were 4.76 MPa, 6.12 MPa, 7.82 MPa, respectively. The results obtained by Abaqus for models with the stacking sequences [±30/90] FW, [±45/90] FW, [±60/90] FW were 4.88 MPa, 5.84 MPa, 8.17 MPa, respectively. Therefore, the correctness of the modeling strategy was validated.

Mesh sensitivity analysis is conducted for composite cylindrical shell, as shown in Figure 2. For the composite cylindrical shell, a total of 25,160 elements are used for the finite element model, taking into account both computational time and accuracy.

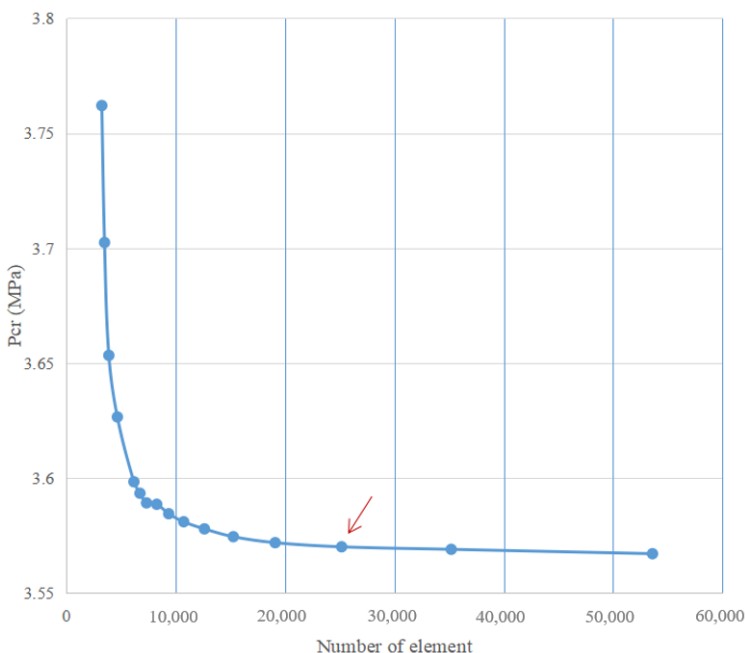

**Figure 2.** Mesh sensitivity analysis for composite cylindrical shell.

*2.2. Finite Element Analysis Results*

Finite element analysis is performed to obtain the buckling mode shape for the composite cylindrical shell subjected to external hydrostatic pressure, as shown in Figure 3. It can be seen that the first buckling mode has three half-waves in the circumferential direction and one half-wave in the axial direction. The critical buckling pressure is 3.6439 MPa.

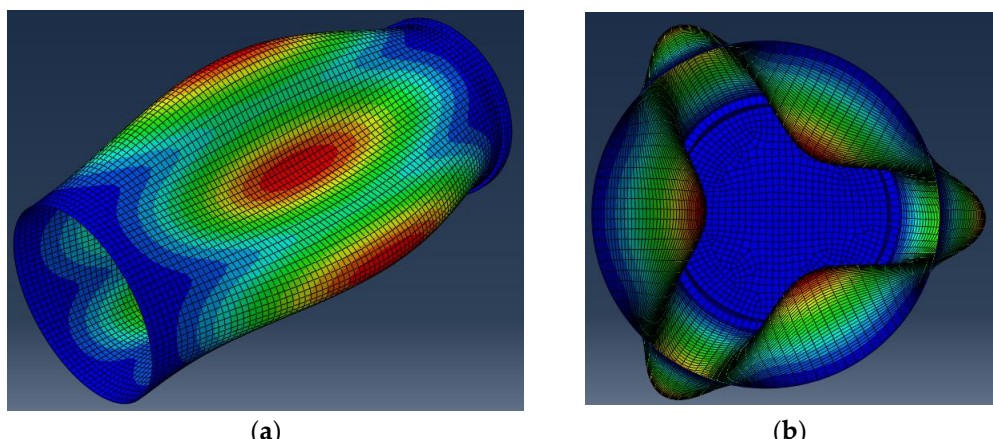

|   (**a**)   |   (**b**)   |

**Figure 3.** Buckling mode shape for composite cylindrical shell: (**a**) front view; (**b**) left view.

## 3. Sparse Polynomial Chaos Expansion

### 3.1. Modeling and Validation

Polynomial chaos expansion (PCE) is a surrogate model that mimics the true input–output relationship of a stochastic system, and the assessment of PCE is much faster than that of finite element analysis. PCE is computationally intensive for high-dimensional problems, and thus sparse PCE is employed instead, which takes into account both the convergence rate and the numbers of models assessed [37–39].

For a stochastic system $Y$ with $M$ independent components $X$ described by the joint probability density function $f_X$, $Y$ has a finite variance; then, the PCE of $Y$ is defined as follows:

$$Y = \sum_{\alpha \in N^M} y_\alpha \psi_\alpha(X) \tag{1}$$

where $y_\alpha$ refers to the expansion coefficients, $\psi_\alpha(X)$ refers to the multivariate polynomials orthonormal with respect to the joint probability density function $f_X$, and $\alpha$ is a $M$ dimensional multi-index that identifies the components of the multivariate polynomials $\psi_\alpha(X)$.

The multivariate polynomials $\psi_\alpha(X)$ are obtained as a product of univariate polynomials $\varphi_{\alpha_i}^{(i)}(x_i)$:

$$\psi_\alpha(X) = \prod_{i=1}^{M} \varphi_{\alpha_i}^{(i)}(x_i) \tag{2}$$

where $\varphi_{\alpha_i}^{(i)}(x_i)$ is the univariate orthogonal polynomial of the $i$th variable with degree $\alpha_i$. Classic polynomials correspond to specific probability distributions; for instance, the Hermite polynomial corresponds to Gaussian distribution, the Legendre polynomial corresponds to uniform distribution, and the Jacobi polynomial corresponds to Gamma distribution.

For practical applications, the sum in Equation (1) needs to be truncated to a finite sum:

$$Y \approx Y^{PCE} = \sum_{\alpha \in A} y_\alpha \psi_\alpha(X) \tag{3}$$

where $A \subset N^M$ is the truncation set of multiple indexes of multivariate polynomials.

After applying the least angle regression algorithm to compute the expansion coefficients, the polynomials that have the greatest effect on the model output are retained and sparse PCE is obtained. Due to the orthonormality of the polynomial basis, the moments of the sparse PCE are encoded in the coefficients. The mean and variance of the sparse PCE can be computed as:

$$\mu^{PCE} = y_0 \tag{4}$$

$$V^{PCE} = \sum_{\alpha \in N^M, \alpha \neq 0} y_\alpha^2 \tag{5}$$

where the mean is the constant $y_0$, and the variance is the sum of squares of all of the coefficients for the polynomials.

First-order Sobol' indices measure the effect of the input variable alone:

$$S_i = \frac{V_{x_i}[E_{x_{-i}}(Y|x_i)]}{V(Y)} \tag{6}$$

where $V$ is the variance and $E$ is the expectation. $x_{-i}$ represents the set of all variables except $x_i$. Total Sobol' indices measure the effect of the input variable as well as higher-order interactions:

$$S_i^T = \frac{E_{x_{-i}}(V_{x_i}(Y|x_i))}{V(Y)} \tag{7}$$

On the basis of the sparse PCE, the first-order Sobol' indices can be obtained as follows:

$$S_i = \frac{\sum\limits_{a \in A_i} y_\alpha^2}{V(Y)} \tag{8}$$

where $A_i = \{a \in N^M, a_i > 0, a_{i \neq j} = 0\}$.

Total Sobol' indices share the same expression as first-order Sobol' indices, except that $A_i = \{a \in N^M, a_i > 0\}$.

According to the general framework of uncertainty quantification proposed by Sudret [40], quantification of the uncertainty of composite cylindrical shells includes three steps, as shown in Figure 4.

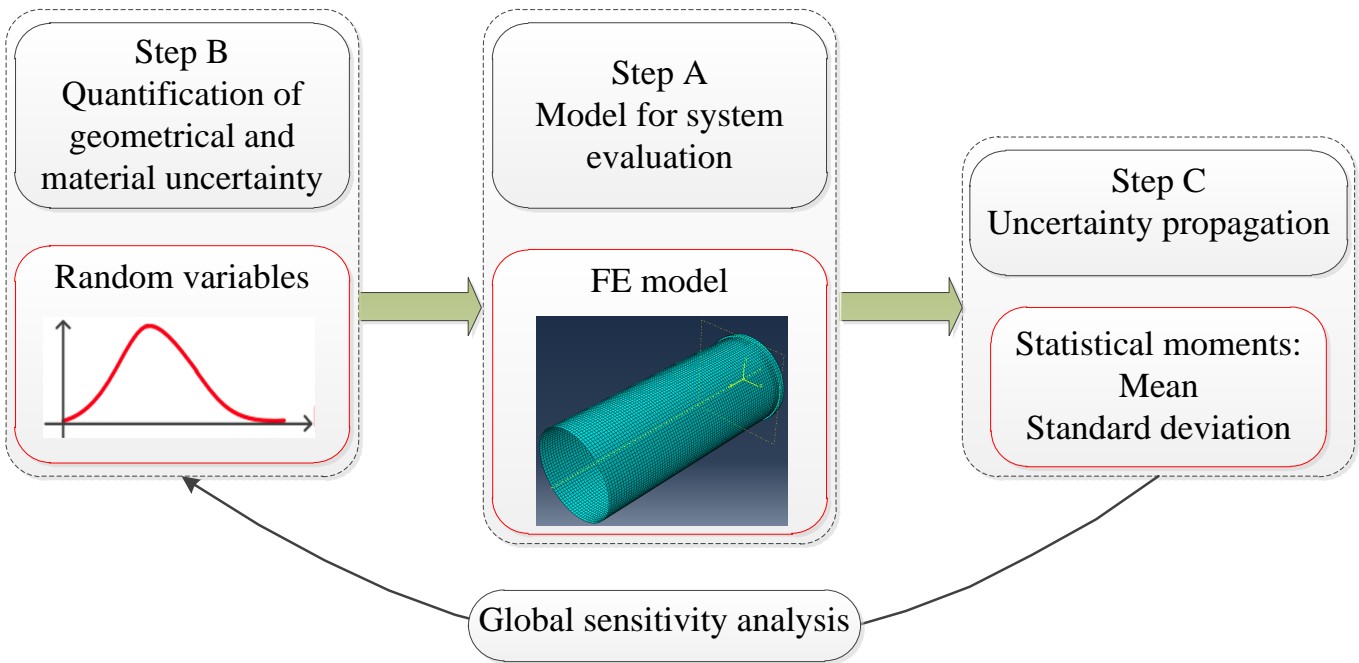

**Figure 4.** General framework for uncertainty quantification.

Step A: The FE model is adopted to obtain the system response or output Y (critical buckling pressure) for the random input parameters X (longitudinal modulus, transverse modulus, shear modulus, Poisson's ratio, orientation angle, and ply thickness).

Step B: As a consequence of the inherent uncertainties of geometrical dimensions and material properties associated with composite materials, the probabilistic method is adopted to model the random variables subjected to a specific probability distribution of the inherent uncertainties.

Step C: The uncertainties with respect to the mechanical properties and geometrical dimensions are propagated to critical buckling pressure. With the aim of evaluating the influence of the uncertainties in the mechanical properties and geometrical dimensions on the critical buckling pressure, uncertainty quantification is conducted and the statistical moments (mean, standard deviation) of critical buckling pressure and global sensitivity analysis results are obtained.

With respect to the data-driven sparse PCE established in this study, design of experiment (DOE) was first conducted, and the samples of random input variables subject to specific probability distributions were obtained and evaluated using FE analysis. Then, the data-driven sparse PCE was constructed and validated with respect to predictive accuracy. Statistical moments and global sensitivity analysis can be obtained instantly using the constructed sparse PCE.

For the data-driven sparse PCE, the material properties of the composite material and the geometrical dimensions were random input variables, while critical buckling pressure determined on the basis of FE analysis was the random output. On the basis of the constructed sparse PCE, uncertainty quantification of was conducted for the composite shell; the flowchart for the data-driven sparse PCE is shown in Figure 5.

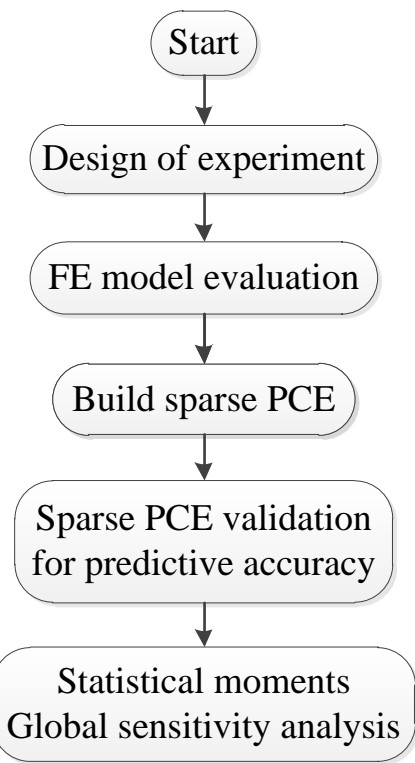

**Figure 5.** The flowchart for the data-driven sparse PCE.

　　As for DOE, the Latin hypercube method was adopted for sampling the random input variables. The sampling data were split into training and testing data sets, with sample sizes of 350 and 100, respectively. Based on the training data set, the data-driven sparse PCE was trained, and it was tested on the testing data set in order to validate its predictive accuracy.

　　The accuracy of the constructed PCE was validated using a testing data set with 100 samples. The critical buckling pressures obtained by means of the finite element method $Y_{FEM}$ and PCE prediction $Y_{PCE}$ obtained using the PCE model are shown in Figure 6. The horizontal axis indicates the value of $Y_{FEM}$, while the vertical axis indicates the value of $Y_{PCE}$. The blue point indicates the predicted critical buckling pressure $Y_{PCE}$ obtained using PCE for a certain $Y_{FEM}$. Nearly all blue points lie near the line y = x, indicating that $Y_{PCE}$ is approximately equal to $Y_{FEM}$ for all of the testing data. This shows good agreement between the finite element method $Y_{FEM}$ and the PCE prediction $Y_{PCE}$; therefore, it can be concluded that the PCE predicts the critical buckling pressure with high accuracy.

　　The normalized histogram of critical buckling pressure obtained using the finite element method $Y_{FEM}$ and PCE prediction $Y_{PCE}$ on the basis of the testing data set is shown in Figure 7. The total area under the normalized histogram is 1. The histogram of critical buckling pressure obtained using the finite element method $Y_{FEM}$ is colored blue, and the histogram obtained using PCE prediction $Y_{PCE}$ on the basis of the PCE model is colored orange. The results indicate good matches.

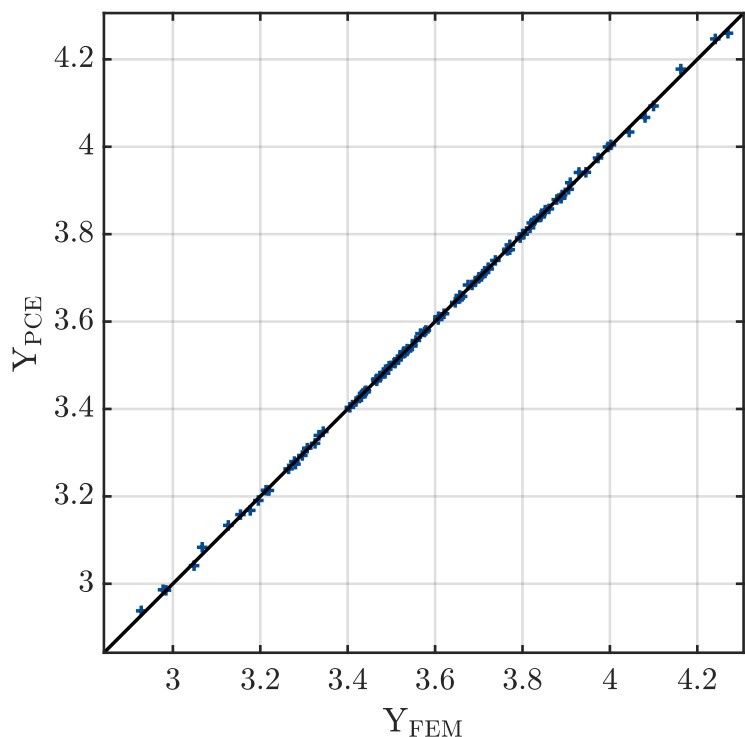

**Figure 6.** Critical buckling pressure obtained by the finite element method $Y_{FEM}$ and PCE prediction $Y_{PCE}$.

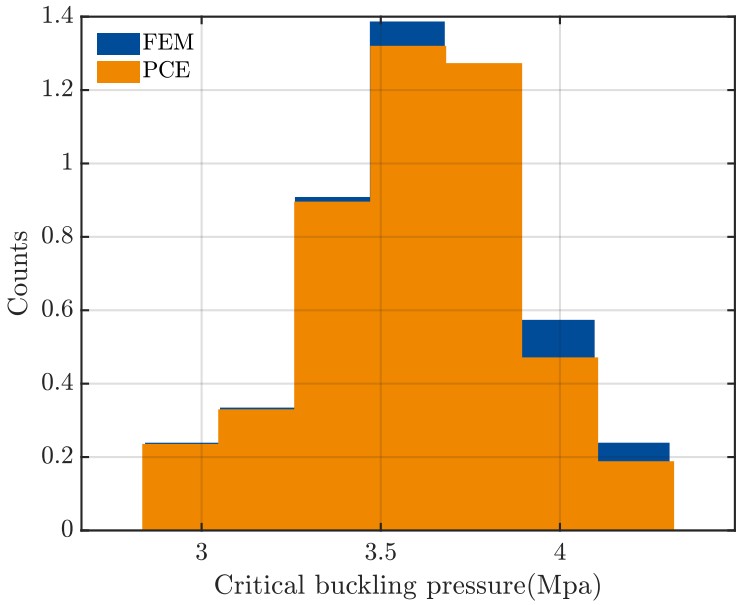

**Figure 7.** Normalized histogram of critical buckling pressure obtained using the finite element method $Y_{FEM}$ and PCE prediction $Y_{PCE}$.

### 3.2. Uncertainty Quantification

As presented in Table 3, the sparse PCE converges at polynomial degree 1. The sparse PCE was trained on the training data set and tested on the testing data set in order to validate its predictive accuracy. The validation error on the testing data set was $2.8282 \times 10^{-4}$. Hence, the constructed PCE was found to be accurate. Moreover, the number of input variables was 29, and the sample size of training data set was 350, which is about 12 times the number of input variables. The sparse PCE was able to effectively

handle the high-dimensional problem with a relatively small data set, while still achieving the desired accuracy. The mean critical buckling pressure was 3.5777 MPa, the standard deviation of critical buckling pressure was 0.3149 MPa, and the coefficient of variation of critical buckling pressure was 8.801%. The critical buckling pressure obtained using deterministic analysis was 3.6439 MPa, and the mean critical buckling pressure obtained using the PCE was 3.5777 MPa, with these results indicating good matches.

**Table 3.** Sparse PCE results.

| Sparse PCE Result | Value |
|---|---|
| Number of input variables | 29 |
| Maximal degree | 1 |
| Size of full basis | 30 |
| Size of sparse basis | 23 |
| Full model evaluations | 350 |
| Leave-one-out error | $2.9839 \times 10^{-4}$ |
| Validation error | $2.8282 \times 10^{-4}$ |
| Mean value | 3.5777 |
| Standard deviation | 0.3149 |
| Coefficient of variation | 8.801% |

Global sensitivity analysis was performed on the basis of analysis using the Morris method and Sobol' indices, respectively; meanwhile, on the basis of the validated sparse PCE, the effects of the material properties and geometrical dimensions on the variations in the critical buckling pressure of composite cylindrical shells were assessed.

On the basis of the total and first-order Sobol' indices, as shown in Figure 8, the first-order Sobol' indices of the random input variables (material properties and geometrical dimensions) were used to measure the marginal variance contribution of individual random variables to variances in the output, depicted in blue. Total Sobol' indices were used to measure the total variance contribution of individual random variables, including the interactions with all of the other random variables, as depicted in orange. E1, namely longitudinal elastic modulus, had the greatest influence on variations in the critical buckling pressure of composite cylindrical shells, with Sobol' indices approaching 1, whereas the other material properties and geometrical dimensions had smaller effects, as their Sobol' indices were close to 0. The first-order Sobol' indices were nearly the same as the total Sobol' indices for all random variables, implying that there were almost no interactions between the random variables. From the subfigure of Figure 8 depicted in red, it can be seen that E2, G12, A2, T1, T2, T3, T4, T5, T6, T7, T8, T9, T10 were the most influential random variables for the variation in output other than E1.

With respect to the Morris method of sensitivity analysis, the greater the value μ* of a random variable, the more sensitive the random variable is. σ represents the interaction between the random variables, and the greater the value σ of a random variable, the greater the interaction occurring among the random variables. The results of the global sensitivity analysis using the Morris method are shown in Figure 9. E1, namely longitudinal elastic modulus, has a relatively high μ* and low σ, whereas the other random variables have low values of both μ* and σ. Hence, E1, namely longitudinal elastic modulus, has the greatest influence on variations in the critical buckling pressure of composite cylindrical shells, whereas the other material properties and geometrical dimensions only have a small effect. All of the random variables have low values of σ, indicating that the interactions among the random variables are small.

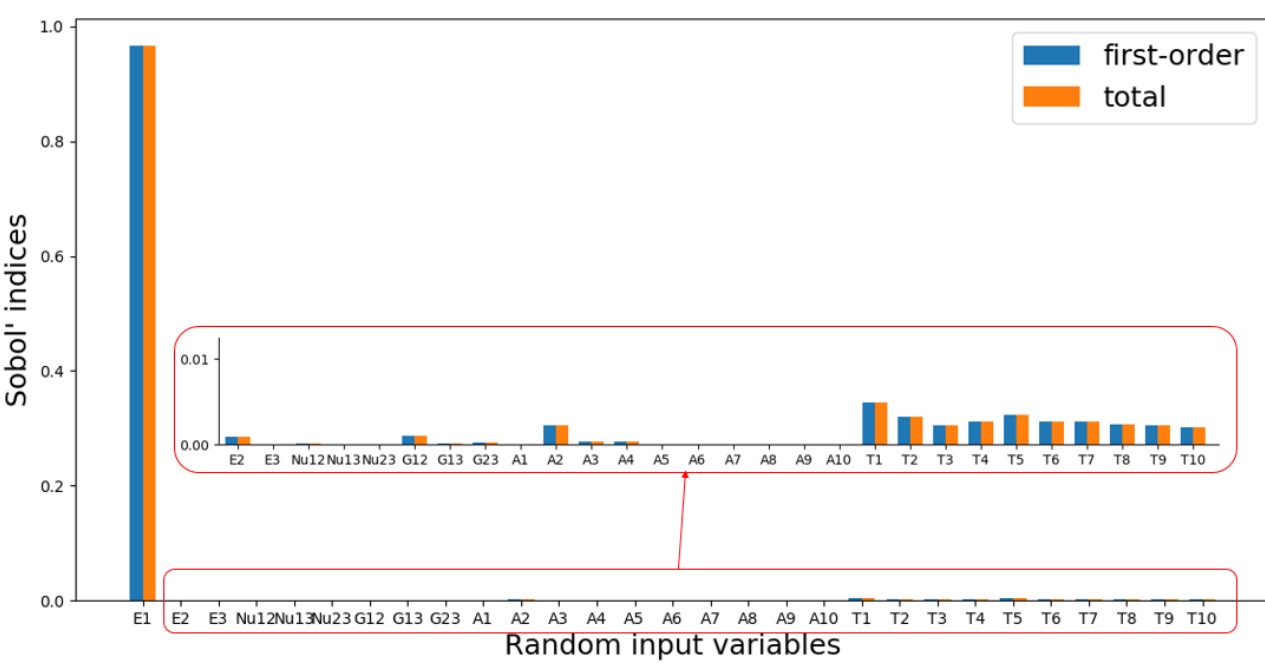

**Figure 8.** The first-order and total Sobol' indices.

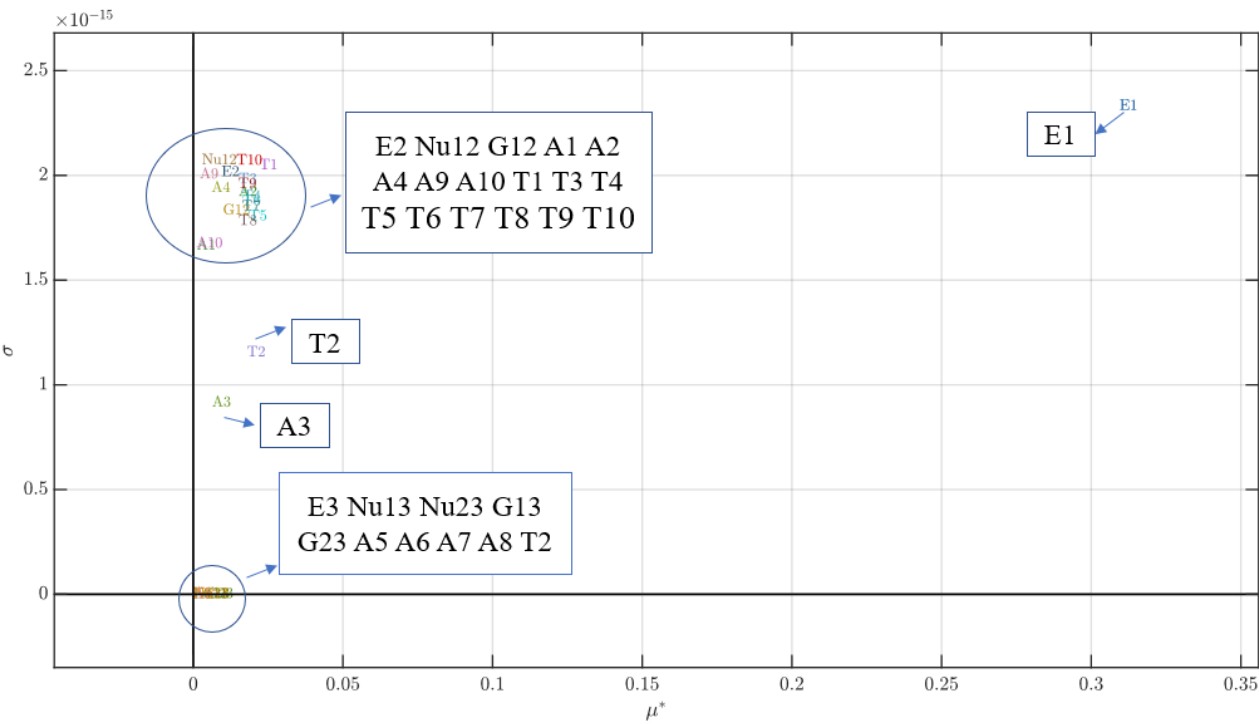

**Figure 9.** Global sensitivity analysis results using the Morris method.

The values of μ* for all random variables except E1 are shown in Figure 10, where it can be seen that E2, G12, A2, T1, T2, T3, T4, T5, T6, T7, T8, T9, T10 are the most influential random variables for the variation in output apart from E1. This is in agreement with the results of the Sobol' indices. The results of g the lobal sensitivity analysis using the Morris method are consistent with those obtained using the Sobol' indices. The correctness of the global sensitivity analysis results is thus validated.

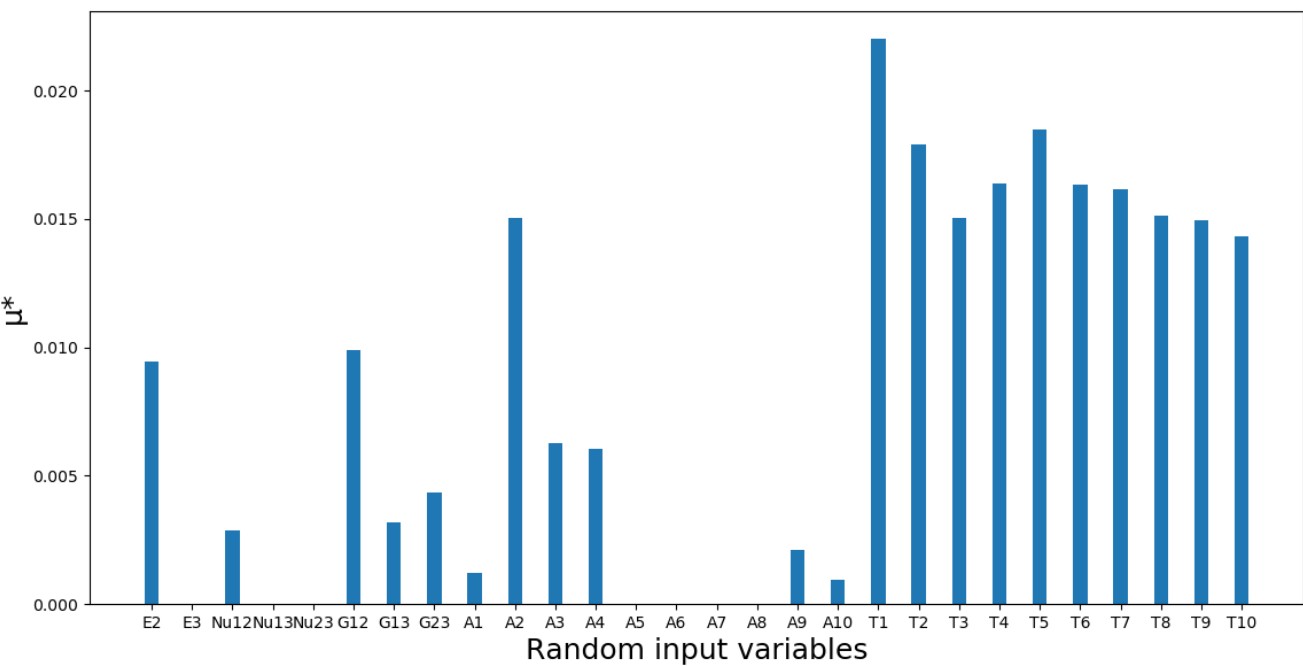

**Figure 10.** Values of μ* for all random variables except E14.

## 4. Conclusions

Traditional methods of uncertainty quantification, such as Monte Carlo simulation and the response surface method, are either time consuming with low convergence rates or unable to deal with high-dimensional problems. This study aims at the quantification of high-dimensional uncertainty for composite cylindrical shells while taking into account uncertainties regarding material properties and geometrical dimensions. Sparse PCE was built and validated with respect to predictive accuracy on the basis of limited design samples, following which the mean and standard deviation of critical buckling pressure and global sensitivity analysis were instantly available. In light of the global sensitivity analysis, the influences of the material and geometrical uncertainty on critical buckling pressure were discussed. Sparse PCE is effective for the quantification of high-dimensional uncertainty in composite cylindrical shells with the desired accuracy. Some crucial conclusions can be drawn, as follows.

1. The uncertainty of the longitudinal modulus of composite materials has a significant influence on the critical buckling pressure of composite cylindrical shells, whereas the uncertainties with respect to the transverse modulus, shear modulus, and Poisson's ratio have a weak influence. When the coefficient of variation of ply thickness and orientation angle does not surpass 2%, the uncertainties regarding ply thickness and orientation angle have a weak influence on the critical buckling pressure.
2. E2, G12, A2, T1, T2, T3, T4, T5, T6, T7, T8, T9, T10 are the most influential random variables on variations in critical buckling pressure other than E1.
3. Sparse PCE is effective for the 29-dimensional problem with a limited design sample consisting of 350 samples (about 12 times the dimensionality), and normalization did not need to be performed for input variables ranging from 0.421 to 121 in this study, indicating the robustness of sparse PCE.
4. The critical buckling pressure obtained using FEM and sparse PCE indicates good matches.

**Author Contributions:** Conceptualization, M.C. and X.Z.; methodology, M.C. and X.Z.; software, M.C.; validation, M.C. and X.Z.; formal analysis, M.C.; investigation, K.S.; resources, M.C.; data curation, M.C.; writing—original draft preparation, M.C.; writing—review and editing, M.C. and X.Z.; visualization, M.C.; supervision, X.Z. and G.P.; project administration, X.Z. and G.P.; funding acquisition, G.P. All authors have read and agreed to the published version of the manuscript.

**Funding:** This work was supported by the National Natural Science Foundation of China under Project Nos. 51979226, 51879220, 6180330, 51909219, and 51709229, the Fundamental Research Funds for the Central Universities, China under Project Nos. 3102019HHZY030019, 3102019JC006, 3102019HHZY03005, and 3102020HHZY030018, and the fellowship of China Postdoctoral Science Foundation (No. 2020M673492).

**Institutional Review Board Statement:** Not applicable.

**Informed Consent Statement:** Not applicable.

**Data Availability Statement:** Not applicable.

**Conflicts of Interest:** The authors declare no conflict of interest.

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
