# Peer review of "Sparse Polynomial Chaos Expansion for Uncertainty Quantification of Composite Cylindrical Shell with Geometrical and Material Uncertainty"

_jmse, doi:10.3390/jmse10050670_

Round 1
Reviewer 1 Report
The manuscript focuses on the uncertainty quantification of critical buckling pressure of composite cylindrical shells with both geometrical and material uncertainty by means of sparse polynomial chaos expansion (PCE). Numerical results are presented and some basic conclusions are drawn.
In its present form, the manuscript cannot be accepted for publication in the journal mainly because Section 3 lacks essential information on the proposed approach. A more detailed description (including the basic equations) of sparse PCE implementation for the computation of the critical buckling pressure (shown in Figure 6) is needed. Moreover, the expressions of total and first order Sobol’ indices used for sensitivity analysis should be provided.
I’m also skeptical about the use of normal distribution for strictly positive material properties, such as the elastic moduli E1, E2, E3 of Table 1. The authors should comment on that.
Finally, Figure 1 could be omitted as it does not provide any additional information with respect to Figure 2.
Reviewer 2 Report
In this study, the sparse polynomial chaos expansion has been employed to investigate the uncertainty of the buckling behavior of composite cylindrical shells. Both geometrical and material uncertainties have been considered in this research. The research is interesting, but its novelty is not adequate and acceptable. however, the following comments can improve the paper:
1- In the abstract, the authors should present more novelties of their work. Only employing a specific method to consider uncertainty is not novel!
2- The authors should improve the literature review by considering some relevant and recent articles, such as:
- Journal of the Brazilian Society of Mechanical Sciences and Engineering 41(10), 419, 2019
- Composite Structures 279, 114798, 2022
- Mechanics of Advanced Materials and Structures 29 (4), 594-612, 2022
- World Journal of Engineering 16 (5), 636-647, 2019
3- The convergence study to achieve the required size of mesh has been missed in this paper. The authors should explain how they select the number of elements used for discretization.
4- The boundary condition of the structure should be presented.
5- Please explain why the whole of the structure has been modeled while its geometry is symmetric.
6- What is the difference between the present research and the following reference:
Journal of Physics: Conference Series. Vol. 2174. No. 1. IOP Publishing, 2022.
7- The English language of the article should be improved grammatically and lexically.
8- On page 8, Line 216, the text and the table have been messed up.
9- The resolution of some figures is not adequate.
10- More explanations are required for figure 8.
11- Nothing in figure 9 is clear.
12- The main contribution and novelty of the present research should be discussed.
13- The procedure of the research should be clarified step by step. Providing a flowchart can be helpful for readers to better understanding.
Round 2
Reviewer 1 Report
My comments have been successfully addressed by the authors and the revised manuscript is recommended for publication in its present form.
Reviewer 2 Report
The authors have answered to the reviewer's comments completely.